# Stool Elastase as an Independent Prognostic Factor in Patients with Pancreatic Head Cancer

**DOI:** 10.3390/jcm11133718

**Published:** 2022-06-27

**Authors:** Honam Hwang, Hongbeom Kim, Hee Ju Sohn, Mirang Lee, Hyeong Seok Kim, Youngmin Han, Wooil Kwon, Jin-Young Jang

**Affiliations:** Departments of Surgery, Seoul National University Hospital, Seoul 03080, Korea; honam9311@gmail.com (H.H.); generalheeju@gmail.com (H.J.S.); effly5026@gmail.com (M.L.); hskim880701@gmail.com (H.S.K.); vickijoa@gmail.com (Y.H.); wildoc78@gmail.com (W.K.); jangjy4@snu.ac.kr (J.-Y.J.)

**Keywords:** stool elastase, pancreatic exocrine insufficiency, malnutrition, pancreatic ductal adenocarcinoma, pancreatic head cancer

## Abstract

(1) Background: Patients with pancreatic exocrine insufficiency (PEI) have an increased risk of malnutrition, which in turn increases morbidity and mortality and is frequent in pancreatic head cancer. This study aimed to analyze the utility of PEI measured using the stool elastase (SE) level to predict the prognosis of patients with pancreatic head cancer. (2) Methods: Patients who underwent pancreaticoduodenectomy for pancreatic cancer at our institution between 2011 and 2015 were included. Only patients with data on preoperative SE levels were analyzed. Patients were classified into low and high SE groups based on preoperative SE levels (low < 100 µg/g < high). (3) Results: The median preoperative SE level was 67.2 µg/g, and 84 of 143 (58.7%) patients were included in the low SE group. The two groups had significantly different overall survival (OS) and disease-free survival (DFS), and the low SE group had a worse prognosis. In multivariate analysis, SE level < 100 µg/g and lymph node metastasis were independent poor prognostic factors for OS and DFS. (4) Discussion: PEI measured using SE levels is an independent prognostic factor in patients with pancreatic head cancer undergoing pancreaticoduodenectomy. Since poor nutritional status may be related to prognosis in patients with low levels of stool elastase preoperatively, aggressive treatment may be required.

## 1. Introduction

Pancreatic cancer is a leading causes of cancer mortality in developed countries and is one of the most lethal malignant neoplasms worldwide [1]. The estimated 5-year survival rate for pancreatic cancer is <5% [1]. The significant prognostic factors include tumor size, lymph node metastasis, surgical margin status, tumor markers, and adjuvant chemotherapy [2]. Pancreatic exocrine insufficiency (PEI) and nutritional factors are also associated with survival in patients with pancreatic cancer [3,4]. Obstruction of the pancreatic duct by tumor growth can lead to pancreatic fibrosis [5]; this causes progressive destruction of functioning pancreatic tissue, resulting in indigestion and malabsorption and ultimately increasing the risk of nutritional deficiencies [6]. In many cases, malnutrition has negative effects on the quality of life and efficacy of tumor therapy, which in turn increases morbidity and mortality [7].

At diagnosis, the prevalence of PEI is 44.5–68.0%, with the presence of a tumor in the pancreatic head region ultimately leading to impaired exocrine function in most patients [8,9]. In particular, pancreatic head cancer is common and associated with a high level of PEI, possibly related to the obstructive effect on the ductal system caused by tumor growth, compared to pancreatic body or tail cancer [10]. There are several methods for diagnosing PEI, including direct and indirect methods. The secretin–pancreozymin test is a direct test to assess pancreatic function, while determination of pancreatic enzymes in the serum and stool is an indirect test to assess pancreatic function [11]. Pancreatic elastase-1 produced by the pancreas is a highly stable enzyme during intestinal transit and can, thus, be measured in stool samples [12,13]. Stool elastase (SE) is a non-invasive and sensitive test for the detection of PEI [11,14]. Therefore, the SE assay has been used widely in clinics as an objective measure of exocrine pancreatic function. This test can help determine a patient’s treatment plan because nutritional status can be easily identified before surgery.

This study aimed to analyze the prognosis of patients with pancreatic head cancer according to PEI measured using SE levels.

## 2. Materials and Methods

### 2.1. Patient Selection and Data Collection

We enrolled patients with pancreatic head cancer who underwent pancreaticoduodenectomy (PD) at Seoul National University of Hospital, Seoul, South Korea between January 2011 and December 2015. Patients who underwent palliative resection and neoadjuvant chemotherapy or who died within 1 month after operation were excluded. Patients who did not have data on preoperative SE levels were excluded.

Demographic and clinical data were also collected. All patients underwent routine laboratory tests, including tumor marker and nutritional tests, such as those for transferrin, prealbumin, albumin, and protein. All patients underwent either PD, pylorus-preserving PD (PPPD), or total pancreatectomy. Postoperative complications after curative resection were classified according to the Clavien–Dindo classification. Complications were defined as CD grade ≥ 3. The surgical specimens were pathologically confirmed and staged according to the American Joint Committee on Cancer 8th edition. Survival was measured from the date of surgery to the date of death or the last follow-up evaluation. This study was approved by the Institutional Review Board of Seoul National University Hospital (SNUH; approval number: 2102-073-1196).

### 2.2. Stool Elastase Level Measurement

PEI was measured using SE levels. Stool specimens were obtained 1–2 days before surgery. SE was measured using an enzyme-linked immunosorbent assay kit (ScheBo Pancreatic Elastase 1 Stool Test, Biotech AG, Giessen, Germany).

In healthy individuals, the concentration of SE is >200 µg/g, and a concentration of <200 µg/g is indicative of PEI [6]. PEI was considered severely reduced if SE levels were <100 µg/g [14,15]. Patients were divided into two groups according to their SE level—(i) low SE group (SE level < 100 µg/g) and (ii) high SE group (SE level ≥ 100 µg/g). The survival outcomes of patients in both groups were also compared.

### 2.3. Statistical Analysis

Continuous variables are presented as mean and standard deviation. Categorical variables are presented as numbers and percentages. In order to assess the normality of the distribution, a Kolmogorov–Smirnov test was introduced. The two groups were compared using the chi-square test or Fisher’s exact test. The 5-year overall survival (5YOS) and 5-year disease-free survival (5YDFS) were estimated according to the Kaplan–Meier method. Multivariate analysis was performed using a Cox regression model to evaluate the significant predictive factors and their relative roles. Multivariate analysis was performed using factors with *p* < 0.1 in univariate analysis. Statistical analyses were performed using SPSS 25.0 for Windows software (SPSS Inc., Chicago, IL, USA). A *p*-value < 0.05 indicates local statistical significance.

## 3. Results

### 3.1. Patient Characteristics

Between January 2011 and December 2015, 327 patients with pancreatic head cancer underwent PD at SNUH. Data on preoperative SE levels were available in 215 patients. In total, 52 patients who underwent palliative resection, 16 patients who underwent neoadjuvant chemotherapy, and 4 patients who died within 1 month were excluded (Figure 1). The mean age of the patients was 64.5 years; 87 patients were males and 56 were females. The median preoperative stool elastase level was 67.2 µg/g, and 111 patients had an SE level of <200 µg/g. Finally, 59 (41.3%) patients were classified in the high SE group (SE level ≥ 100 µg/g) and 84 (58.7%) were included in the low SE group (SE level < 100 µg/g), indicating severe PEI.

There was no significant difference between the groups in terms of basic demographics. Nutritional parameters, such as transferrin, prealbumin, protein, and albumin levels, were not significantly different between the two groups. Among the postoperative clinical factors, the grade of postoperative pancreatic fistula showed a clinical relationship with SE levels (Table 1). However, there were no differences in preoperative carbohydrate antigen (CA) 19-9 (*p* = 0.501), surgical margin status (*p* = 0.212), T-stage (*p* = 0.114), N-stage (*p* = 0.296), and adjuvant chemotherapy (*p* = 0.358) between the groups.

### 3.2. Survival Analysis According to the Stool Elastase Group

The low SE group had significantly shorter overall survival (OS) and disease-free survival (DFS) than the high SE group (Figure 2). The patients with SE levels < 100 µg/g was 16.7%, while the 5YOS rate of patients with SE levels ≥100 µg/g was 32.7% (median 5YOS: 17 vs. 25 months, *p* = 0.035). In addition, the 5YDFS rate in the low SE group was 11.9% and that in the high SE group was 25.0% (median 5YDFS: 8 vs. 14 months, *p* = 0.006).

### 3.3. Prognostic Factors for Survival

Univariate analysis revealed several significant prognostic factors for OS and DFS (*p* < 0.05). ASA (American society of anesthesiologists) score (*p* = 0.095), preoperative CA 19-9 level (*p* = 0.009), preoperative SE level (*p* = 0.040), and *n*-stage (*p* = 0.001) (Table 2) were prognostic factors of OS. Similarly, ASA score (*p* = 0.064), preoperative CA 19-9 level (*p* = 0.002), preoperative SE level (*p* = 0.008), and *n*-stage (*p* = 0.001) were prognostic factors for DFS (Table 3).

Multivariate analysis showed that SE level (hazard ratio [HR] 1.487, *p* = 0.048) and *n*-stage (HR 1.852, *p* = 0.005) were significantly correlated with OS (Table 2). In addition, SE level (HR 1.894, *p* = 0.003) and *n*-stage (HR 1.605, *p* = 0.014) were significantly correlated with DFS (Table 3). For both OS and DFS, SE level was found to be a significant prognostic factor.

## 4. Discussion

PEI is a major contributor to malnutrition in pancreatic cancer and is associated with poor prognosis, and the incidence of PEI is higher in pancreatic head cancer than in pancreatic body or tail cancer [10,16,17]. However, few studies have analyzed PEI using SE levels only in pancreatic head cancer. In the present study, including patients undergoing curative resection for pancreatic head cancer, we examined whether PEI measured using SE levels is a prognostic factor for pancreatic head cancer and revealed that it was an independent prognostic factor.

PEI can be measured using several methods, including the direct hormone-stimulated pancreatic function test, secretin endoscopic pancreatic function test, and a 72-h stool fat test [18,19,20]. However, the SE assay is the most widely accepted method for the detection of PEI because of its convenience and sensitivity [11,14]. This test uses <1 g of stool, is not affected by dietary status, and can be performed even after 14 days of refrigerated storage [21]. Therefore, since the SE assay is the most widely used diagnostic method for PEI in clinics, our study defined PEI and severe PEI based on SE levels.

Several studies have analyzed the SE level and prognosis in pancreatic cancer. Partelli et al. reported that an extremely reduced SE level (<20 µg/g) is an independent predictor of survival in advanced pancreatic cancer [4]. However, this study, which included patients with stage III or IV pancreatic cancer and the reference SE level, obtained ambiguous results. Lim et al. reported that a low SE level (≤200 µg/g) was a significant prognostic factor for recurrence of pancreatic cancer in patients who underwent curative resection involving any location of the pancreas [22]. As mentioned earlier, PEI depends on tumor location [10]. Because the surgical method varies depending on the tumor location and nutritional status may vary accordingly, analysis is necessary in patients undergoing surgery with the same surgical method. Our study included patients with pancreatic head cancer in whom the surgical method was unified with PD and was set based on an SE level of 100 µg/g, which indicates severe PEI. The SE assay shares independently established but equivalent reference intervals of <100, 100–200, and >200 µg/g for severe pancreatic insufficiency, moderate insufficiency, and normal pancreatic function, respectively [23].

Generally, the diagnostic criterion for PEI is an SE level of 200 µg/g. However, when the study was conducted based on an SE level of 200 µg/g, patients with SE levels < 200 µg/g accounted for 77.6% patients, with only 23.4% patients classified into the low SE group; hence, statistical analysis was difficult because of the large difference in distribution between the two groups. Therefore, we performed a study based on an SE level of 100 µg/g, defining any value above this as severe PEI. We further performed the study with cutoff values of 50 µg/g and 200 µg/g, obtaining similar results with 50 µg/g. The 5YOS rate of patients with SE levels < 50 µg/g was 14.1% and that of patients with SE levels ≥ 50 µg/g was 30.0% (median 5YOS: 17 vs. 26 months, *p* = 0.010). In addition, 5YDFS rate of patients with SE levels < 50 µg/g was 9.4% and that of patients with SE levels ≥ 50 µg/g was 20.3% (median 5YDFS: 8 vs. 13 months, *p* = 0.006). Multivariate analysis also showed that SE levels were significantly correlated with OS (HR 1.592, *p* = 0.016) and DFS (HR 1.596, *p* = 0.012).

The survival of patients with pancreatic cancer remains very poor, as it is usually diagnosed at advanced stages. Many investigators have used multivariate analysis to identify prognostic markers in patients undergoing pancreatic resection. The prognostic factors associated with pancreatic cancer include age, tumor size, surgical margin status, pathologic grading of differentiation, preoperative CA 19-9 level, blood loss, and postoperative adjuvant therapy [24,25]. In this study, other prognostic factors that could affect pancreatic cancer, such as age, preoperative CA 19-9, surgical margin status, stage, and adjuvant chemotherapy were comparable between the two groups. Cox multivariate analysis of the present study revealed that stool elastase level and *n* stage were independent prognostic factors. Unusually, adjuvant chemotherapy did not show any statistically significant differences in this study. This is probably because the majority (85%) of patients received adjuvant chemotherapy.

There have been studies comparing nutritional prognostic factors, although none for PEI. Kanta et al. reported that the prognostic nutrition index (PNI) is associated with overall survival and postoperative complications and is an independent prognostic factor of poor survival in patients with pancreatic cancer [3]. In addition, Kim et al. reported that preoperative malnourished patients experience poor clinical outcomes after pancreaticoduodenectomy [16]. The fact that nutritional status appears to be a useful predictor of postoperative outcome does not mean that the reversal of preoperative malnutrition improves outcomes. Nonetheless, appropriate nutritional assessment and aggressive nutritional management for patients undergoing major surgery are recommended in the European Society for Parenteral and Enteral Nutrition Guidelines on Parenteral Nutrition [26].

Because PEI can be caused by an obstruction of the pancreatic duct caused by tumor growth, it is likely that the size of the tumor is associated with PEI. In the present study, there was no statistically significant difference, but the low SE group had more T stages 3 and 4 compared to the high SE group. The poor prognosis of pancreatic cancer is multifactorial, but PEI plays an important role. Nutritional deficiencies due to obstruction of the pancreatic duct and a decrease in residual pancreatic function after surgery may result in decreased immunity and nutritional status. As a result, there are limitations in the selection of adjuvant chemotherapy regimens and doses, and tolerance to chemotherapy is reduced [27]. Consequently, nutritional interventions such as perioperative nutritional support and pancreatic enzyme replacement therapy can be helpful in determining prognosis [28]. Therefore, it is important to recognize and treat pancreatic cancer patients with PEI preoperatively, which is expected to have a low survival rate, although it is not known if an aggressive nutritional support may improve survival beyond the improvement of the nutritional status.

In this study, the grade of POPF (postoperative pancreatic fistula) was statistically different between the two groups. Interestingly, pancreatic fistula occurred infrequently in patients in the low SE group. The occurrence of POPF depends on several factors, such as the hardness of the pancreas, type of anastomosis, and use of a pancreatic stent [29]. Kanda et al. reported that the risk of pancreatic fistula decreased in patients with a low BMI (body mass index). It was postulated that lean patients often lose weight owing to tumor growth, and the anastomotic procedure is performed easily because of reduced fat tissue surrounding the pancreas. In addition, tumor growth and concomitant pancreatitis, which involves sclerotylosis of the pancreatic duct and a firmer pancreatic texture, are associated with a decreased risk of POPF [3].

The study was based on a prospectively collected database, but it has the fundamental limitation of retrospective studies. We tried to enroll all consecutive patients; however, we had to exclude patients without data on SE levels. However, despite these limitations, many patients were analyzed as homogenous patient groups for a short period of time.

In conclusion, PEI measured using SE levels was an independent prognostic factor for patients with pancreatic head cancer who underwent PD. Therefore, in patients with preoperative low stool elastase levels, poor nutritional status may be relevant to prognosis, so adequate treatment, such as aggressive chemotherapy, nutritional support, and a short-term follow-up strategy, may be required.

## Figures and Tables

**Figure 1 jcm-11-03718-f001:**
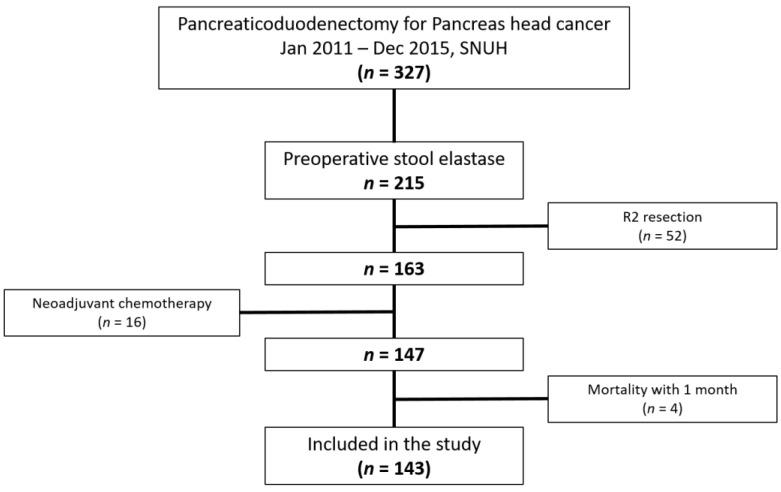
Flowchart of patient selection.

**Figure 2 jcm-11-03718-f002:**
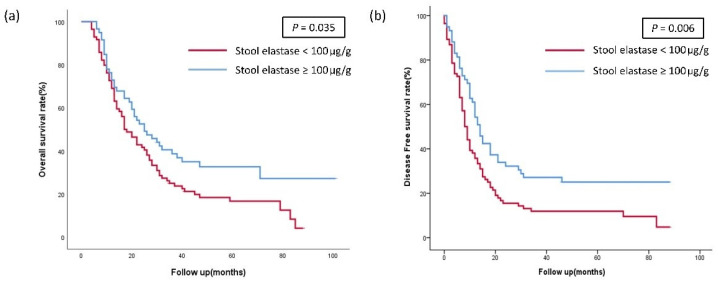
(**a**) The overall survival (OS) and (**b**) disease-free survival (DFS) of the patients according to the stool elastase group.

**Table 1 jcm-11-03718-t001:** Clinicopathological features according to preoperative stool elastase levels.

Variables	Total (*n* = 143)	Low SE Group <100 µg/g (*n* = 84)	High SE Group ≥100 µg/g (*n* = 59)	*p*-Value
Sex (Male/Female) (%)	87 (60.8)/56 (39.2)	53 (63.1)/31 (36.9)	34 (57.6)/25 (42.4)	0.602
Age (mean ± SD, years)	64.5 ± 9.4	63.8 ± 9.1	65.5 ± 9.9	0.345
BMI (mean ± SD, kg/m^2^)	22.6 ± 2.8	22.6 ± 2.6	22.6 ± 3.1	0.461
ASA class (%)				0.708
I II III	30 (21.0) 01 (70.6) 12 (8.4)	16 (19.0) 60 (71.4) 8 (9.5)	14 (23.7) 41 (69.5) 4 (6.8)	
DM, Yes (%)	58 (40.6)	37 (44.0)	21 (35.6)	0.387
Smoking, Yes (%)	24 (16.8)	15 (17.9)	9 (15.3)	0.821
Pre-op CA19-9, U/mL (mean ± SD)	901 ± 2669	913 ± 2397	884 ± 3037	0.501
Pre-op transferrin, mg/dL (mean ± SD)	234 ± 43	232 ± 41	236 ± 46	0.438
Pre-op prealbumin, mg/dL (mean ± SD)	23.8 ± 7.3	23.1 ± 7.3	24.8 ± 7.4	0.681
Pre-op protein, g/dL (mean ± SD)	6.9 ± 0.6	6.9 ± 0.6	6.9 ± 0.5	0.415
Pre-op albumin, g/dL (mean ± SD)	3.9 ± 0.4	3.9 ± 0.4	3.9 ± 0.4	0.406
Postoperative hospital day (mean ± SD, day)	16.0 ± 9.6	15.0 ± 7.8	17.4 ± 11.6	0.449
R0 resection status (%)	124 (86.7)	70 (83.3)	54 (91.5)	0.212
Complication CD grade ≥ 3 (%)	22 (15.4)	9 (10.7)	13 (22.0)	0.098
POPF (%)				<0.001
No Biochemical leak Grade B	103 (84.6) 29 (20.3) 11 (7.7)	72 (85.7) 9 (10.7) 3 (3.6)	31 (52.5) 20 (33.9) 9 (13.6)	
T stage (%)				0.114
1 2 3 4	21 (14.7) 97 (67.8) 22 (15.4) 3 (2.1)	14 (16.7) 51 (60.7) 16 (19.0) 3 (3.6)	7 (11.9) 46 (78.0) 6 (10.2) 0 (0.0)	
*n* stage (%)				0.296
Negative Positive	55 (38.5) 88 (61.5)	29 (34.5) 55 (65.5)	26 (44.1) 33 (55.9)	
Adjuvant chemotherapy (%)	121 (84.6)	69 (82.1)	52 (88.1)	0.358
Adjuvant radiotherapy (%)	81 (56.6)	47 (56.0)	34 (57.6)	0.865

SE, Stool Elastase; SD, Standard Deviation; BMI, Body Mass Index; ASA, American Society of Anesthesiologists; DM, Diabetes Mellitus; CD, Clavien–Dindo; POPF, Postoperative Pancreatic Fistula.

**Table 2 jcm-11-03718-t002:** Univariate and multivariate analysis of risk factors for overall survival.

Variable	Univariate			Multivariate		
	HR	95% CI	*p*-Value	HR	95% CI	*p*-Value
Sex, (male) vs. female	0.995	0.680–1.455	0.979			
Age, years (≤65) vs. >65	1.053	0.873–1.269	0.590			
BMI, kg/m^2^ (≤23) vs. >23	1.035	0.858–1.250	0.719			
ASA class (I) II III	1.443 0.915	1.024–2.034 0.687–1.219	0.095 0.036 0.545	1.405 0.869	0.989–1.996 0.667–1.202	0.121 0.058 0.463
Pre-op CA 19–9, U/mL (≤37) vs. >37	1.772	1.151–2.727	0.009	1.495	0.958–2.332	0.076
Stool elastase, µg/g (≥100) vs. <100	1.501	1.019–2.211	0.040	1.487	1.003–2.206	0.048
T stage (T1, T2) vs. (T3, T4)	1.011	0.621–1.645	0.965			
*n* stage (Negative) vs. Positive	2.098	1.404–3.134	<0.001	1.852	1.210–2.835	0.005
Adjuvant Chemotherapy (No) vs. Yes	1.111	0.858–1.439	0.424			
Adjuvant Radiotherapy (No) vs. Yes	1.086	0.900–1.311	0.389			

HR, Hazard Ratio; CI, Confidential Interval; BMI, Body Mass Index; ASA, American Society of Anesthesiologists; CA 19-9, Carbohydrate Antigen 19-9.

**Table 3 jcm-11-03718-t003:** Univariate and multivariate analysis of risk factors for disease-free survival.

Variable	Univariate			Multivariate		
	HR	95% CI	*p*-Value	HR	95% CI	*p*-Value
Sex, (male) vs. female	1.026	0.708–1.487	0.891			
Age, years (≤65) vs. >65	1.107	0.772–1.588	0.580			
BMI, kg/m^2^ (≤23) vs. >23	0.969	0.675–1.391	0.864			
ASA class (I) II III	1.491 0.890	0.744–2.991 0.473–1.675	0.064 0.260 0.718	1.353 0.790	0.565–2.790 0.407–1.534	0.050 0.413 0.486
Pre-op CA 19-9, U/mL (≤37) vs. >37	1.925	1.269–2.923	0.002	1.525	0.980–2.373	0.061
Stool elastase, µg/g (≥100) vs. <100	1.651	1.137–2.399	0.008	1.894	1.238–2.895	0.003
T stage (T1, T2) vs. (T3, T4)	1.280	0.798–2.054	0.305			
*n* stage (Negative) vs. Positive	2.089	1.417–3.079	<0.001	1.605	1.103–2.337	0.014
Adjuvant Chemotherapy (No) vs. Yes	1.049	0.628–1.754	0.855			
Adjuvant Radiotherapy (No) vs. Yes	1.028	0.857–1.233	0.767			

HR, Hazard Ratio; CI, Confidential Interval; BMI, Body Mass Index; ASA, American Society of Anesthesiologists; CA 19-9, Carbohydrate Antigen 19-9.

## Data Availability

Not applicable.

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
