# Peer review of "Stool Elastase as an Independent Prognostic Factor in Patients with Pancreatic Head Cancer"

_jcm, 2022, doi:10.3390/jcm11133718_

Round 1
Reviewer 1 Report
Dear Editor
Thanks for inviting me to review the manuscript jcm-1778091 entitled "Stool elastase as an independent prognostic factor in patients with pancreatic head cancer".
The title is novel enough, and the study methodology seems correct. My minor comments to the authors are as follow.
Abstract:
- "Patients were 13 classified into low and high SE groups based on preoperative SE levels (low < 100 µg/g < high)" What about the SE=100? Please include it in high level category.
- The second sentence of conclusion is not clear. " In such patients, management of severe PEI, such as nutritional support, is required." In which patients (which category)?
Also, this conclusion is not based on the results. Did the authors evaluate the effect of nutritional support?
Introduction:
- Please mention the importance of using such biomarkers to evaluate the patients' survival. Why this results is important for the literature?
Statistical analysis:
- Where did you use the median and IQR/mean and SD? Did you evaluate the normality of the distributions? By which test?
Results:
- Figure two is not necessary.
Discussion:
- Please discuss the importance of such biomarkers to evaluate the patients' survival.
Conclusion:
- Please revise the conclusion according to the results. For example, which patients needs further attention? Which patients has more probability of having poor prognosis? etc.
Regards
Reviewer 2 Report
Hwang et al. demonstrated that stool elastase might be a predictor of the mortality of patients with pancreatic malignancy. The pancreatic malignancy can influence the exocrine function, in turn, pancreatic elastase production. The manuscript is well written except for the conclusion; we should carefully interpret the results because the present study did not show the effect of nutritional support in those with pancreatic malignancy, and just indicated the association between stool elastase and mortality of the patient.
